Reviving lost shadows: investigating the habitat ecology of the rediscovered hispid hare (Caprolagus hispidus) in Nepal

Prasai Aakriti 1
Dhami Bijaya bdhami@ualberta.ca 2
Saini Apoorv 2
Thapa Roshna 1
Samant Kopila 3
Regmi Krishika 4
Dhami Rabin Singh 5
Sadadev Bipana Maiya 6
Adhikari Hari hari.adhikari@helsinki.fi 7 8
1 College of Natural Resource Management, Faculty of Forestry, Agriculture and Forestry University , Katari, Udayapur , Nepal
2 Department of Biological Sciences, University of Alberta , Edmonton , Canada
3 Faculty of Forestry, Agriculture and Forestry University , Hetauda , Nepal
4 Institute of Forestry, Pokhara Campus, Tribhuvan University , Pokhara , Nepal
5 Institute of Forestry, Hetauda Campus, Tribhuvan University , Hetauda , Nepal
6 Natural Resources and Environmental Studies, University of Northern British Columbia , Prince George , BC , Canada
7 Department of Geosciences and Geography, University of Helsinki , Helsinki , Finland
8 Forest Nepal , Butwal , Nepal
Żyła Dagmara
Electronic publication date: 2024 Sep 26
Publication date: 2024
Volume: 12
Electronic Location ID: e18034
Received 2024 Jun 5; Accepted 2024 Aug 12
Copyright: ©2024 Prasai et al.
Copyright year: 2024
Copyright holder: Prasai et al.
License: This is an open access article distributed under the terms of the Creative Commons Attribution License, which permits unrestricted use, distribution, reproduction and adaptation in any medium and for any purpose provided that it is properly attributed. For attribution, the original author(s), title, publication source (PeerJ) and either DOI or URL of the article must be cited.
License URL: https://creativecommons.org/licenses/by/4.0/

Keywords: Conservation strategies, Dynamic nature, Endangered, Grassland, Lagomorph

Funding: The APC was funded by the Helsinki University Library. The funders had no role in study design, data collection and analysis, decision to publish, or preparation of the manuscript.

==============================
The endangered hispid hare (Caprolagus hispidus) is one of the least studied mammal species. The recent rediscovery of hispid hare in Chitwan National Park (CNP) after three decades, necessitated urgent conservation measures. A detailed investigation into the species’ ecology is imperative for developing evidence-based conservation strategies to support these efforts. The main objective of this study is to investigate the current distribution pattern and habitat preferences, offering vital insights for the effective preservation and management of the species and its habitat. Between 28 January and 13 February 2023, fifty-two surveys using strip transects were carried out in the potential areas. If any indirect signs of the presence of the hispid hare were identified, the corresponding value is coded as 1 “used plot”. In contrast, a value of 0 was assigned if there is an absence of any indirect signs “habitat availability plot”. Nine habitat predictors (habitat type, ground cover, distance to water source, distance to roads/path/firelines, ground condition, dominant plant species, presence/absence of (anthropogenic disturbance, predators, and invasive species), were measured from both plot types (“used plot” and “habitat availability plot”). Our research indicates a clumped distribution pattern within the CNP, with the Sukhibhar grassland identified as a key hotspot. Our study reveals the hispid hare’s adaptability to diverse grassland conditions, favoring both tall and short grasslands. It is essential to integrate the species’ preference for various grassland habitats and critical water sources to enhance conservation strategies. This requires a comprehensive grassland management plan for Chitwan National Park that preserves habitat diversity, safeguards key water sources, and adapts to evolving environmental conditions.

Introduction

Lagomorpha, a globally distributed order within the superorder Euarchontoglires, includes rodents, lagomorphs, tree shrews, colugos, and primates, found on all continents except Antarctica (Murphy et al., 2001; Forsyth et al., 2005; Chapman & Flux, 2008; Scott et al., 2014; Burgin et al., 2018; Smith et al., 2018). Lagomorphs are herbivores and are classified into two extant families: Ochotonidae (pikas) and Leporidae (rabbits and hares) (Smith et al., 2018). Currently, there are 12 recognized genera within Lagomorpha, encompassing 108 species (Burgin et al., 2018; Kraatz et al., 2021), with a considerable number falling under the umbrella of threatened status (Vulnerable, Endangered, and Critically Endangered) (IUCN, 2024) and/or are also identified as Evolutionary Distinct and Globally Endangered (EDGE) species (Fontanesi et al., 2016). Nepal, for instance, is home to the world’s rarest mammal with a monotypic genus—the hispid hare (Caprolagus hispidus) (Dhami et al., 2023a). Hispid hare is globally listed as endangered by the International Union for Conservation of Nature (IUCN) Red List of Threatened Species (Aryal & Yadav, 2019) and classified under Appendix I of the Convention on International Trade in Endangered Species of Wild Flora and Fauna (CITES) (CITES, 2021). In Nepal, this species is designated as a protected priority species under the National Park and Wildlife Conservation Act of 1973 (GoN, 1973) and is recognized as endangered in the National Red List series (Jnawali et al., 2011; Dhami et al., 2023a).

The scope of knowledge on the distribution of the hispid hare has always been limited (Dhami et al., 2023a; Sadadev et al., 2021). Historically, hispid hare species once thrived in the foothills of the southern Himalayas, encompassing territories from Uttar Pradesh through southern Nepal, the northern reaches of West Bengal extending to Assam, and into Bangladesh as far south as Dacca (Aryal & Yadav, 2019). In the mid-1960s, ecologists hypothesized the potential extinction of the hispid hare; however, the capture of a live specimen in 1971 within the Barnadi Wildlife Sanctuary, Assam, conclusively confirmed its survival (Dhami et al., 2023a). Currently, the species displays a sporadic presence across southern Asia, spanning Nepal, Bhutan, Bangladesh, and India (Nath & Machary, 2015; Khadka et al., 2017; Dhami et al., 2023a), existing within an elevation range of 100–250 m (Aryal & Yadav, 2019). In recent reports, it has been documented that hispid hare inhabit isolated pockets within Shuklaphanta National Park (ShNP), Bardia National Park (BNP), and Chitwan National Park (CNP) in Nepal (Khadka et al., 2017; Sadadev et al., 2021).

The hispid hare primarily occupies floodplain or alluvial grasslands characterized by early successional tall grasses, commonly called “elephant grass” (Bell, Oliver & Ghose, 1990). Tall grasslands may function as a habitat refugia in later stages of ecological succession by forming an understory structure, particularly close to rivers or within forest clearings, as well as in areas abandoned after cultivation or village settlements (Chapman & Flux, 1990; Dhami et al., 2023a). During the dry months of the year (from November to April), intentional fires are ignited in the grassland and adjacent forests as a strategy to regulate the faunal composition of the area (Sadadev et al., 2021; Dhami et al., 2023b). In response, hispid hare relocate to marshy areas or grasses adjacent to riverbanks that are not susceptible to burning to seek refuge (Aryal & Yadav, 2019). As the monsoon attains its peak, the thatch becomes waterlogged, moving hispid hare towards forested areas (Ghose, 1978). The hispid hare predominantly selects thatch shoots and roots as its dietary preference, employing a feeding behavior characterized by biting off at the base and removing outer sheaths before consumption (Oliver, 1980). Despite the critical role played by the Terai floodplain grasslands in supporting biodiversity and fulfilling local communities’ needs through fuel and thatch roofing materials (Bhatta, 1999; Sadadev et al., 2021), these areas face imminent threats. The primary challenges include natural succession, excessive grazing by cattle, unregulated harvesting of thatch, and uncontrolled burning (Sadadev et al., 2021; Dhami et al., 2023a; Dhami et al., 2023b). Consequently, suitable habitat refuges for small mammals, including the hispid hare, are significantly reduced in these grasslands (Sadadev et al., 2021).

Since the 1970s, Nepal has demonstrated remarkable progress in conservation efforts by strategically developing and expanding its network of protected areas (PAs), encompassing more than 20% of the nation’s total land area (Heinen et al., 2019). The commitment to conservation is further underscored by the implementation of rigorous laws aimed at safeguarding both endangered species and crucial natural habitats (Dhami et al., 2023c). This steadfast dedication has resulted in a multitude of successes. While substantial attention in zoological research has understandably gravitated towards high-profile large mammals, driven by funding availability and policy priorities, significant knowledge gaps persist for numerous other vertebrate taxa, particularly in the realms of ecological study and small mammal inventory, including lagomorph species (Chand, Khanal & Chalise, 2017; Khadka et al., 2017; Nidup, 2018; Dhami et al., 2023a; Dhami et al., 2023c). Following the establishment of CNP, the first protected area of Nepal, Nepal’s government (Department of National Parks and Wildlife Conservation) has persistently directed its efforts towards conducting comprehensive presence/absence surveys (employing camera traps, transect surveys, etc.) on the lowland protected areas of Nepal, with a strategic emphasis on megafauna (Khadka et al., 2017; Dhami et al., 2023b). Amidst these ongoing conservation efforts, an interesting turn of events unfolded. On 30 January 2016, within CNP, researchers led by Khadka et al. (2017) rediscovered an individual hispid hare after three decades. This surprising discovery occurred during a survey of grassland birds, including the Bengal florican (Houbaropsis bengalensis). The rediscovery prompted conservation managers to take immediate and targeted actions for the hispid hare and its habitat, necessitating a reassessment of existing management strategies and the formulation of specific plans to address the needs of this newly rediscovered threatened species. Consequently, meticulous investigation has become imperative to accurately map the current distribution of the hispid hare and understand the key characteristics of its habitat. Thus, the primary objective is to assess the hispid hare’s current distribution patterns and habitat preferences in CNP, offering vital insights for effectively preserving and managing the species and its habitat. These critical findings are essential for shaping effective conservation policies and represent a forward-looking approach to ensure the enduring survival of the hispid hare.

Materials and Methods

Study area

The research was conducted in Chitwan National Park (CNP), the first national park in Nepal and a designated UNESCO World Heritage site (see Fig. 1). CNP, along with its buffer zone, is situated in the southern part of Central Nepal, spanning across Chitwan, Nawalparasi, Parsa, and Makwanpur districts. The park’s coordinates range from N27°20′19″ to 27°43′16″ longitude and E83°44′50″ to 84°45′03″ latitude, covering an expanse of 952.6 km2 in the Rapti Valley within the Siwalik physiographic region. The buffer zone, extending from N27°28′23″to 27°70′38″ longitude and E83°83′98″ to 84°77′38″ latitude, encompasses an area of 729.37 km2. CNP’s elevation varies from 100 m in river valleys to 815 m in the Siwalik hills (Bhuju et al., 2007). The park is rich in biodiversity, hosting an impressive array of wildlife, including almost 68 species of mammals, over 576 species of birds, 49 species of reptiles and amphibians, 120 species of fishes, and numerous invertebrates, all of which play a crucial role in the park’s ecological processes (CNP, 2015). Among the charismatic species, Chitwan National Park is home to 128 Royal Bengal Tigers, the highest number among all other national parks (Pokharel, 2022). Moreover, the park is home to the second-largest population of the Greater One-horned Rhinoceros in the subcontinent, ranking just below Kaziranga National Park in India (Mandal, 2022). The dominant grass species in the grasslands include Saccharum munja, Saccharum bengalense, and Imperata cylindrica (Khadka et al., 2017).

Figure 1 The spatial distribution of hispid hare in Chitwan National Park, where distinct areas of concentration are visually emphasized by yellow circles.

Data collection

Before an extensive field survey, a preliminary key informant survey was conducted in January 2023. The participants included representatives from the national park (specifically rangers, n = 3), members of the buffer zone management committee (n = 2), and nature guides (n = 3). The key informants were chosen based on their prior experience working in the national park. To assist them in the process, detailed historical distribution maps (color-printed) were provided during interviews. Following this, the research team conducted an initial field trip to locations recommended by the key informants, aiming to understand the terrain and plan for the comprehensive field survey. Researchers recorded GPS locations using handheld Garmin devices whenever indirect evidence of the hispid hare, such as pellets, dens, or grass cutting, was encountered.

The extensive field visit was conducted during the cold winter, specifically from 28 January to 13 February 2023. This temporal preference was guided by the sparser vegetation cover in the tropic region in this season (Safford, 2004; Sanusi et al., 2013; Neupane et al., 2022). The spare vegetation cover enhances the chances of spotting hispid hare pellets, thus mitigating potential biases linked to undetectability issues (Sadadev et al., 2021; Dhami et al., 2023a; Dhami et al., 2023b). Non-intrusive molecular methods serve as more compelling evidence for confirming species’ presence (Buglione et al., 2020b; Buglione et al., 2020a). However, since the hispid hare is nocturnal and elusive, direct sightings are rare, making pellet analysis a practical alternative for confirming its presence. The size and shape of pellets are recognized as reliable indicators for species identification (Aryal et al., 2012; Chand, Khanal & Chalise, 2017). Hispid hare pellets can be distinguished from those of the rufous-tailed Indian hare (Lepus nigricollis ruficaudautus) by their unique larger size and flattened, rounded shape (Dhami et al., 2023a). In contrast, Indian hare pellets are generally smaller, darker, and exhibit an elliptical form with a pointed end (Refer to Sadadev et al., 2021 for photographic evidence).

We used the transect method for our survey, laying down a long linear transect of 500 m. Adjacent transects were spaced 200 m apart to minimize extraneous variation within the location and minimize the possible overlapping of pellets (Dhami et al., 2023a). We placed alternate strip transects along this main transect, each 20 m long and 2 m wide (Sadadev et al., 2021). These strip transects were positioned at 100-meter intervals along the main transect, as shown in Fig. 2. A team led by the first author and experienced field guides with a background in wildlife research and monitoring systematically searched along the strip transect for indirect signs of the hare’s presence (pellets, dens, and grass cutting). This method was used because direct sightings were difficult due to the hare’s nocturnal and elusive nature (Bell, 1987; Bell, Oliver & Ghose, 1990). Whenever the indirect signs of a hispid hare were encountered, a circular plot of radius 1.78 m was established with detected signs at the center (Gyawali, 2003). Subsequently, another circle with the same radius was established in a random direction, 100 m away from the center of the initial circle with detected signs (Khulal et al., 2021; Neupane, Chhetri & Dhami, 2021; Neupane et al., 2022). These circular plots served as standard habitat samples, independent of whether the hispid hare was present. If any signs of a hispid hare were detected within the circles, a value of 1 (“utilized plots”) was assigned; otherwise, if no evidence was found, a value of 0 (“habitat availability plot”) was assigned.

Figure 2 Illustration outlining the layout of planned transects and plots designed for assessing the habitat characteristics of hispid hares in specific sampling units.

Alternating rectangular boxes denote strip transects and circles signify sampling plots, referred to as ‘used plots’ or ‘Availability plots’. These plots were used to record different habitat parameters. A coin flip was used to select the first transect, and subsequent transects were carried out alternatively.

Based on the literature review (Aryal et al., 2012; Chand, Khanal & Chalise, 2017; Dhami et al., 2023a; Khadka et al., 2017; Nath & Machary, 2015; Oliver, 1984; Rastogi, Raj & Chauhan, 2020; Sadadev et al., 2021; Tandan et al., 2013) and initial on-site survey we recorded nine predictor variables that could potentially influence the habitat preferences of hispid hare across all circles of both types (“used plots” and “availability plots”) (Table 1). Firstly, the water shapefile was extracted from DEM (Digital elevation model) with a resolution of 12.5 m and a Landsat Image 8 (USGS, 2021), and the Road/Path/Fireline shapefile was extracted from the Open street map (OSM, 2021). Then, the distance to the nearest water source and Road/Path/Fireline was calculated using a “Nearer function” in ArcGIS 10.8. version for the presence locations (geographic coordinates) of the hispid hare. Habitat-type categorization and recording of the presence/absence of anthropogenic disturbance, invasive species, and Predator was based on personal observations (Koirala et al., 2020; Khulal et al., 2021). For determining the presence of predators such as Felis chaus and Canis aureus, we relied on identifying their fresh signs, specifically pugmarks and scats that were estimated to be 1–3 days old (Khatoon et al., 2019). Similarly, we visually approximated ground cover using a circular bamboo frame (radius: 1.78 m) intersected by two diagonal sticks (Roshetko et al., 2002). We estimated the total percentage of grass species coverage by adding the proportion of coverage in each quadrant within the circular frame. The dominant plant species were identified by determining the species with the highest percentage coverage recorded within the circle (Dhami et al., 2023a). The ground condition was evaluated by considering soil moisture’s presence or absence (Table 1).

Table 1 Detailed information on explanatory variables utilized in the logistic regression model.

Variable	Variable type	Variable category	Values	Data source	
Presence or absence of hispid hare	Dependent	Categorial	• Presence = 1
• Absence = 0	Field survey	
Ground cover % (GCo)	Predictors	Categorial	• Low (0–25%) = 1
• Moderate (26–50%) = 2
• High (51–75%) = 3
• Dense (76–100%) = 4	Field survey	
Habitat type (HT)		Categorial	• Tall grassland (grass > 2 m height) = 1
• Short grassland (grass 25 cm to 2 m in height) = 2
• Open grassland (grass < 25 cm
in height) = 3
• Forest (Dominated by trees of any species) = 4	Dhami et al. (2023a)	
Nearest distance to the water source (m) (WD)		Continuous	Range (10–200)	Field survey	
Ground condition (GC)		Categorial	• Wet (sticky consistency) = 1
• Dry (dusty consistency) = 2	Field survey	
Dominant plant species (DS)		Categorial	•Narenga porphyrocoma = 1
•Saccharum spontaneum = 2
•Imperata cylindrical = 3
•Themeda arundinacea = 4	Dhami et al. (2023b)	
Nearest Road/path/fire distance (m) (RD)		Continuous	Range (20–700)	Field survey	
Presence/absence of anthropogenic disturbance (plastics, jeep safari, controlled or uncontrolled burning, grass cutting) (P.A_AD)		Categorial	• Presence = 1
• Absence = 0	Field survey	
Presence/absence of Predator (scat, pugmark) (P.A_P)		Categorial	• Presence = 1
• Absence = 0	Field survey	
Presence/absence of invasive species (P.A_INV)		Categorial	• Presence = 1
• Absence = 0	Field survey	

Data analysis

The distribution pattern of the hispid hare was first determined using the variance-to-mean ratio (S2/a). The calculation involves S2 = 1n ∑x−a2, where x denotes the number of pellet groups (signs) per sampling unit, a is the mean of the x values, and n represents the number of sampling units (Odum, 1971).

If (S2/a) = 1, i.e., a random distribution,

If (S2/a) < 1, i.e., a uniform distribution,

If (S2/a) > 1, i.e., a clump distribution.

Then, a test for multicollinearity was conducted on the selected independent habitat predictors using the variance inflation factor (VIF) with the “Faraway” package (Boomsma, 2014) in R x 64 4.0.3 (R Core Team, 2020). All selected habitat predictors were included in the final analysis, as none exhibited multicollinearity, with tolerance values exceeding 0.1 and VIF values below 10 (Bowerman & O’connell, 1990). Generalized linear models (GLMs) utilizing a binary distribution were employed to assess the probability of detecting hispid hare with distinct habitat predictors outlined in Table 1. Among the nine habitat predictors, the nearest distance to a water source (WD) and Road/path/firelines were treated as a continuous variable. In contrast, the remaining predictors were considered categorical variables in the analysis. We employed the ‘dredge’ function within the “MuMIn” package (Barton, 2009) to execute the global model, generating all potential models. These models were subsequently ranked using Akaike’s information criterion with correction (AICc) for small samples, with the most effective or dominant models identified by their lower AICc values (Barton & Barton, 2020). The final model-averaged coefficients were obtained by averaging the top candidate models (with delta AIC ≤ 2) (Burnham & Anderson, 2001). This averaging process was performed using the “AICmodavg” package (Mazerolle & Mazerolle, 2017). The predictive efficacy of the best-fitting model was assessed using the area under the curve (AUC) of the receiver operating characteristics (ROC) values ranging from 1 to 0, using the R package “ROCR” (Sing et al., 2005) (Fig. 3). Discrimination performance was considered acceptable for values between 0.7 and 0.8, excellent for values between 0.8 and 0.9, and superior for values exceeding 0.9 (Hosmer, Lemeshow & Lemeshow, 2000).

Figure 3 Illustrating the receiver operating characteristic (ROC) curve for optimal binary logistic regression model assessment, providing a comprehensive performance evaluation through accuracy (ACC) and area under the curve (AUC) metrics.

Results

Distribution pattern of hispid hare

Hispid hare presence was predominantly recorded in Sukhibhar grassland (68.5%), followed by Reu Khola Phanta I (17.5%) and Reu Khola Phanta II (14%) (Fig. 1). The overall distribution exhibited a clumped pattern with a variance to mean ratio >1, i.e., 2.36.

Influencing variables and probability of hispid hare occurrence

Among the binary logistic regression models predicting the likelihood of detecting the hispid hare, the model that includes the combined effects of habitat type and distance to water emerged as the most suitable. This model achieved the lowest Akaike Information Criterion (AIC) value of 160.43 and the highest model weight of 0.36, indicating superior performance compared to all other models (Table 2). Further, the best-fit model demonstrated exceptional performance in explaining hispid hare detection probability, with a 0.803 (80.3%) receiver operating characteristic (ROC) curve area and an impressive 0.805 (80.5%) accuracy (Fig. 3).

Table 2 Akaike Information Criterion (AIC) values and model weights for binary logistic regression models, revealing the most parsimonious and optimal-fit model in predicting the likelihood of hispid hare detection.

Models	d.f	logLik	AICc	ΔAIC	Weight	
Detection∼(HT + WD)	5	−75.01	160.43	0.00	0.36	
Detection∼(HT + RD + WD)	6	−74.20	160.96	0.53	0.27	
Detection∼(HT + P.A_P + WD)	6	−74.38	161.32	0.89	0.23	
Detection∼(HT + P.A_P + RD + WD)	7	−73.74	162.25	1.82	0.14	
Notes.

HT, Habitat type; WD, Water distance; RD, Road distance; P.A_P, Presence/absence of associated fauna; and d.f, Degree of freedom.

Among the nine habitat predictors studied, tall grassland habitat type [HT]1 (β = 1.6009531, S.E = 0.7732176, P = 0.038405), short grassland habitat type [HT]2 (β = 2.6622712, S.E = 0.7349409, P = 0.000292) and water distance [WD] (β = −0.0402213, S.E = 0.0094737, P = 2.18e−05) showed significant links to hispid hare detection (Table 3). The model suggests that an increase in tall and short grassland positively correlates with the likelihood of hispid hare detection. Conversely, as water distance increases, the detectability of hispid hare decreases, and vice versa (Table 3 and Fig. 4).

Table 3 Averaged coefficients of predictor variables and their impact on hispid hare detection in Chitwan National Park.

Variables with significant influence on hispid hare habitat use (Pr (>—z—) < 0.05) are indicated by asterisks (*).

Predictors	Estimate	Std. Error	z value	Pr(>—z—)	
Intercept	7.9412668	1.8332357	4.332	1.48e−05***	
Factor [HT (tall grassland)]	2.6622712	0.7349409	3.622	0.000292***	
Factor [HT (short grassland)]	1.6009531	0.7732176	2.071	0.038405*	
Factor [HT (open grassland)]	0.0529864	0.9678526	0.055	0.956341	
WD	−0.0402213	0.0094737	4.246	2.18e−05***	
RD	0.0003373	0.0006724	0.502	0.615912	
Factor [(P.A_P)1]	0.1729168	0.3536395	0.489	0.624868	

Figure 4 Illustrating the probability of hispid hare detection with water distance.

Discussion

The hispid hare (Caprolagus hispidus), native to Nepal and northern India’s Terai grasslands, confronts significant conservation challenges, largely due to insufficient attention from conservation managers. This study focuses on its current spatial distribution pattern and factors associated with its habitat use in CNP after its rediscovery in 2016. Given habitat changes and increasing human disturbance, the findings of this study hold paramount significance for the formulation and implementation of efficacious and targeted conservation strategies.

Spatial distribution pattern of hispid hare

Until the 1980s, the hispid hare was historically documented in three protected areas in Nepal, namely CNP, ShNP, and BNP (Oliver, 1980). By the 1980s, recorded sightings of the species were limited to BNP and ShNP, raising concerns about its distribution and population status (Tandan et al., 2013). Surprisingly, an individual hispid hare was rediscovered in Sukhibhar grassland within CNP on 30 January 2016, during a targeted survey focused on grassland birds, particularly the Bengal florican (Houbaropsis bengalensis) (Khadka et al., 2017). The rediscovery of the hispid hare in Sukhibhar grassland within CNP offers a beacon of hope for conservationists. Once thought to have expired from CNP, this revelation suggests a broader habitat range than initially recognized, enhancing our understanding of the species. This underscores the significance of thorough and targeted surveys in conservation efforts, encouraging collaborative efforts across protected areas to ensure the long-term survival of this elusive species. In our study, Sukhibhar grassland stands out as a central hub for distributing the hispid hare, followed by Reu Khola Phanta I and Reu Khola Phanta II. Our results are consistent with the findings of Aryal et al. (2012), Chand, Khanal & Chalise (2017), and Dhami et al. (2023a), providing further evidence that the hispid hare species favors the grassland system as an integral part of its habitat. The choice of grasslands is driven by factors such as the availability of food, shelter, and a preferred microhabitat for survival and reproduction (Dhami et al., 2023a). The hispid hare’s strong affinity for grassland ecosystems is a key factor contributing to its clumped distribution. Sadadev et al. (2021) found a clumped distribution of hispid hare in both burned and unburned plots within ShNP, mirroring the results of our study. These results shine a light on the important factor that species with specific habitat requirements often exhibit non-random distribution patterns (Underwood, 1978). Furthermore, CNP harbors the highest number of one-horned rhinoceros (Rhinoceros unicornis) (n = 694) in Nepal, a grassland-dependent species with similar feeding preferences as hispid hare (Subedi et al., 2017). Thus, a study focusing on species interaction is recommended to better understand the dynamics and dependencies within the ecosystem.

Influencing variables and probability of hispid hare occurrence

We observed a positive correlation between the likelihood of detecting hispid hare and the presence of both tall and short grasslands. Aryal et al. (2012) also found that the hispid hare predominantly used tall grasslands (2–6 m) before and after a fire event. Additionally, pellets were discovered in open areas with short grass cover (<2 m), with a notable increase after the fire. This suggests a flexible habitat hispid hare use, highlighting their adaptability to varying grassland conditions. The observed patterns in habitat preference may be influenced by factors such as food availability, shelter, and microhabitat requirements (Bell, 1987; Sadadev et al., 2021; Dhami et al., 2023a). Numerous global studies on hare species, such as the European brown hare (Lepus europaeus), have similarly indicated that grasslands and fallow lands serve as primary habitats for hare (Smith et al., 2004; Sliwinski et al., 2019). This strong habitat preference may stem from the fact that tall grasslands and vegetation act as resting places (Smith et al., 2004; Sliwinski et al., 2019), offering essential shelter and cover against predators (Tapper & Barnes, 1986; Jennings et al., 2006). This protective environment can enhance reproductive success by reducing the risk of predation on offspring (Aryal et al., 2012; Sadadev et al., 2021). Furthermore, the presence of the hispid hare was notably high in habitats dominated by species such as Saccharum spontaneum and Imperata cylindrica (Yadav, 2006; Dhami et al., 2023a), both of which were identified as common components of the hispid hare’s diet in a study by Aryal et al. (2012) conducted in ShNP. During the monsoon season in Assam, India, hispid hare are noted to inhabit forested areas due to waterlogging of the grasslands (Ghose, 1978). However, there is no reported evidence of similar behavior in Nepal (Aryal et al., 2012; Tandan et al., 2013; Chand, Khanal & Chalise, 2017; Sadadev et al., 2021; Dhami et al., 2023a).

After the establishment and growth of Nepal’s protected areas network, many have served as tourist attractions for many years (Heinen & Thapa, 1988), contributing millions of dollars annually to both the national and local economies (Baral et al., 2020). The rise in tourism has resulted in the disruption of wildlife activities, increased stress levels, and the loss or modification of habitats. For instance, (Cosgriff, 1997) observed alterations in the behavior of greater one-horned rhinoceros in response to tourism activities, with animals avoiding typical habitats and seeking refuge in isolated locations within CNP. Similarly, Dhami et al. (2023a) demonstrated a negative correlation between the probability of hispid hare detection and anthropogenic disturbance in ShNP, indicating the direct impact of human activities on the well-being of the hispid hare population and other wildlife.

A negative correlation is evident between the distance from water sources and the detection of hispid hare, indicating that the likelihood of detecting hispid hare decreases as the distance from the water source increases. This aligns with what was found in a prior investigation by Scharine et al. (2011), which documented comparable trends in other lagomorph species. Tandan et al. (2013) also, reported that the hispid hare exhibits a broad distribution, particularly in areas closer to water sources. This could be attributed to the winter season, where water scarcity might compel the hispid hare to remain in areas close to water sources for hydration. Additionally, water bodies often sustain abundant vegetation, attracting various plant species and thereby offering ample foraging opportunities during the dry season (Neupane, Chhetri & Dhami, 2021; Regmi et al., 2022). In contrast to our findings, Nath & Machary (2015) reported a high concentration of hispid hare pellet piles in areas characterized by dry ground conditions and at a distance from water sources. Consistent with this, the presence of hispid hare signs was more commonly found in areas distant from water sources after a fire, especially when there was new growth grass, compared to situations with only unburned grass (Aryal et al., 2012). This suggests that the availability of new grass, characterized by its increased water content, allows hispid hare to extend their habitat beyond water sources following fires (Aryal et al., 2012; Yadav et al., 2008).

Conclusion and conservation implications

In conclusion, our research on the hispid hare’s spatial distribution and habitat preferences within Chitwan National Park (CNP) provides valuable insights for conservation policies. It underscores the importance of continued research on the lagomorph species. Our study reveals a clumped distribution pattern within CNP, with Sukhibhar grassland emerging as a critical hub for the species. Our findings emphasize the adaptability of the hispid hare to various grassland conditions, showcasing a preference for both tall and short grasslands. This flexibility in habitat use suggests that conservation efforts should consider the dynamic nature of the species and the importance of maintaining diverse grassland ecosystems. Furthermore, our research identifies water distance as a crucial factor influencing the likelihood of detecting hispid hare. The negative correlation indicates that proximity to water sources increases the chances of detection. This information is vital for conservation planners, as it sheds light on the species’ seasonal behavior, particularly during the dry winter months when access to water becomes a determining factor in habitat selection. In summary, our research contributes to understanding the hispid hare’s ecology and provides actionable insights for conservation management in Chitwan National Park. As the rediscovery of the hispid hare within CNP sparked renewed conservation efforts, our study reinforces the importance of comprehensive surveys and ongoing monitoring to inform adaptive management strategies. By addressing the identified influencing variables and understanding the species’ habitat preferences, we can develop targeted conservation plans that safeguard the hispid hare and contribute to the overall biodiversity conservation in Nepal’s Terai grasslands.

We gratefully acknowledge the Department of National Park and Wildlife Conservation for providing staff to help with data collection.

Additional Information and Declarations

Competing Interests

Author Contributions

Data Availability

The authors declare there are no competing interests.

Aakriti Prasai conceived and designed the experiments, performed the experiments, analyzed the data, prepared figures and/or tables, authored or reviewed drafts of the article, and approved the final draft.

Bijaya Dhami conceived and designed the experiments, analyzed the data, prepared figures and/or tables, authored or reviewed drafts of the article, and approved the final draft.

Apoorv Saini performed the experiments, analyzed the data, authored or reviewed drafts of the article, and approved the final draft.

Roshna Thapa performed the experiments, analyzed the data, authored or reviewed drafts of the article, and approved the final draft.

Kopila Samant performed the experiments, analyzed the data, prepared figures and/or tables, authored or reviewed drafts of the article, and approved the final draft.

Krishika Regmi performed the experiments, analyzed the data, prepared figures and/or tables, authored or reviewed drafts of the article, and approved the final draft.

Rabin Singh Dhami performed the experiments, analyzed the data, prepared figures and/or tables, authored or reviewed drafts of the article, and approved the final draft.

Bipana Maiya Sadadev performed the experiments, analyzed the data, authored or reviewed drafts of the article, and approved the final draft.

Hari Adhikari conceived and designed the experiments, analyzed the data, prepared figures and/or tables, authored or reviewed drafts of the article, and approved the final draft.

The following information was supplied regarding data availability:

The data is available at Zenodo: Prasai, A. (2024). Reviving Lost Shadows: Investigating the habitat ecology of the rediscovered hispid hare (Caprolagus hispidus) in Nepal [Data set]. Zenodo. https://doi.org/10.5281/zenodo.11413147.

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
