# Peer review of "Reviving lost shadows: investigating the habitat ecology of the rediscovered hispid hare (Caprolagus hispidus) in Nepal"

_PeerJ, doi:10.7717/peerj.18034_

## Round 0.1 · original submission · Major Revisions

Please, address all the reviewers' comments and suggestions. There were concerns regarding the novelty of the study and species identification, which should be solved.

Reviewer 1 ·

Basic reporting

I did not find any novelty in this study.

Experimental design

Similar as published before.

Validity of the findings

No additional findings

Additional comments

Repetitive work

Reviewer 2 ·

Basic reporting

The study deals with the presence and habitat usage of the hispid hare (Caprolagus hispidus). The subject of the study, the said hispid hare is a declining (the IUCN status: Endangered) leporid species, rarely studied and poorly known. Regarding the topic, the study is certainly worth noticing, but there are some issues which should be clarified first to be sure the Authors indeed studied the hispid hare not some other leporid species or even other small animals.

Introduction: Current species count for lagomorphs is closer to 100 (see e.g., Kratz et al. [2021] Lagomorpha as a Model Morphological System. Front. Ecol. Evol. 9:636402. doi: 10.3389/fevo.2021.636402 for a good starting point on general information on the order).
The analysis of spatial distribution of the studied animal seems sound and well executed.

The paper is overall interesting and concerns very important for biodiversity problem, the existence of Caprolagus hispidus, an endangered leporid species. The paper is well organized and I have no major comments to the methods as they are, but I would like to see more explanation on leporid itself, especially its evidence, possible seeings, identification of the pellets etc. That would certainly make paper clearer.

Experimental design

It is not obvious in the study how the animals or in fact the traces of animal's activity were identified. The Authors do not actually mention seeing the animals on the spot and identifying it visually, just studying the places of their supposed activity. The fecal pellets are mentioned but without any specifics. For example, could authors provide information on how to distinguish and identify Caprolagus pellets? Or was any DNA analysis performed to identify the species?

Validity of the findings

I would like to see some kind of explanation on what grounds Authors determine that it was Caprolagus, not any other leporid? Are there any sympatric leporid species in this area? Or was any DNA from pellets analyzed? A short info in Methods chapter or a paragraph included in the discussion would be very welcome.

Additional comments

My additional concerns are with the literature cited. Many entries are in fact not published, but some rather obscure reports (e.g., Ghose 1978) or BSc theses deposited (or not?) at the local libraries; the access to such studies may be potentially problematic and the data included hard to reach or verify. Authors sometimes give scanty information on the papers cited (lack of DOIs, lack of page numbers (e.g., Khadka et al. 2017). I would like to see also the current IUCN List version cited: IUCN. 2024. The IUCN Red List of Threatened Species. Version 2024-1. https://www.iucnredlist.org (I wonder why the 2017 version is cited and accessed in 2017?). In Smith 2018 ‘JHU’ should be spelled ‘Johns Hopkins University”. Overall, this section should be carefully checked for accuracy.

Reviewer 3 ·

Basic reporting

This study is a significant contribution to the understanding of the elusive Hispid Hare's distribution and habitat use in Chitwan. It is commendable that this research is addressing these aspects for the first
time in this region, which holds substantial ecological importance. The study's design is robust and well-thought-out, providing a solid foundation for the research.

While the study provides valuable insights, the outcomes are not particularly novel. The results are very similar to Dhami's work conducted in 2023, which also reported similar findings. The manuscript
requires significant improvement in terms of language and writing quality. The writing is poor, with long and complex sentences that hinder clarity. I recommend a thorough review and editing process
to simplify and streamline the text, ensuring the content is easily understandable and professionally presented.

Experimental design

However, the analysis section requires further attention, particularly concerning the Variance Inflation Factor (VIF) and model building. The VIF values indicate a high level of multicollinearity among the predictor variables, which must be addressed to ensure reliable and interpretable results. I recommend revisiting the model to mitigate multicollinearity issues. The definition of habitat types within the study can be further refined.

Validity of the findings

Currently, habitat type is nested within all the models, which might obscure important variations. A clearer and more distinct categorization of habitat types will enhance the accuracy and relevance of the model
outcomes.

Annotated reviews are not available for download in order to protect the identity of reviewers who chose to remain anonymous.

---

## Round 0.2 · accepted · Accept

I am satisfied with the changes and explanations provided by the authors. In my opinion, the manuscript is ready for publication.